# Spatiotemporal Thermal Control Effects on Thermal Grill Illusion

**DOI:** 10.3390/s23010414

**Published:** 2022-12-30

**Authors:** Satoshi Saga, Ryotaro Kimoto, Kaede Kaguchi

**Affiliations:** Faculty of Advanced Science and Technology, Kumamoto University, Kumamoto 860-8555, Japan

**Keywords:** thermal grill illusion, pre-warming and pre-cooling, spatiotemporal control

## Abstract

The thermal grill illusion induces a pain sensation under a spatial display of warmth and coolness of approximately 40 °C; and 20 °C. To realize virtual pain display more universally during the virtual reality experience, we proposed a spatiotemporal control method to realize a variable thermal grill illusion and evaluated the effect of the method. First, we examined whether there was a change in the period until pain occurred due to the spatial temperature distribution of pre-warming and pre-cooling and verified whether the period until pain occurred became shorter as the temperature difference between pre-warming and pre-cooling increased. Next, we examined the effect of the number of grids on the illusion and verified the following facts. In terms of the pain area, the larger the thermal area, the larger the pain area. In terms of the magnitude of the pain, the larger the thermal area, the greater the magnitude of the sensation of pain.

## 1. Introduction

With the spread of Head Mounted Display (HMD), Virtual Reality (VR) technology is becoming more familiar, and VR technology is developing. Currently, the main developments and research on haptic stimuli are vibrations. However, there are other pain sensations in the human tactile sensation. Therefore, in this study, we proposed employing a spatiotemporal display method to control the thermal grill illusion (TGI). With the method, the pain sensation can be presented more universally during the VR experience.

This illusion induces a burning sensation under a spatial display of warmth and coolness of approximately 40 °C and 20 °C. The sensation was observed by Thunberg in 1896 [1]. The presented warm and cold stimulus temperatures are safe for humans. In this temperature range, the warm receptors (TRPV3, TRPV4) and the cool receptors (TRPM8) work, although neither of them causes a pain sensation [2,3]. Studies vary with regard to the frequency of painful and non-painful paradoxical sensations obtained, and one possible reason for the discrepancy is the experimental paradigm. The ’classic’ combination of 20 and 40 °C [4,5,6] and the combination of 15 and 45 °C [7] produced a non-painful heat sensation. However, 20 and 40 °C induced painful heat elsewhere [8].

In previous research, Stevens et al. reported that temperature perception is based on spatial weighting in thermosensation and that the wider the stimulus range, the stronger the warming sensation that occurs and that doubling the stimulus range half the temperature threshold [9]. Green [10] reported that the phenomenon of referral and domination occurred when they presented thermal stimulation to the index, middle, and ring fingers of subjects at the same time with hot and cold stimuli. In the thermal referral phenomenon, even though the middle finger touches a room-temperature object, the middle finger is perceived as warm when the index and ring fingers are presented with a warm stimulus at the same time. On the other hand, in the domination phenomenon, even when a cold stimulus is presented to the middle finger under similar conditions, it is perceived as ‘warm’.

Hsin-Ni et al., reported detailed results of the thermal referral phenomenon. They examined the perception of the intensity of the sensation resulting from thermal referral to human participants. They found that the sensation was uniform between the three fingers, but its apparent intensity was always lower than the physical intensity applied to the outer two fingers [11]. They considered the reason for the effect to be the link between thermal sensation and tactile somatosensory perception. Furthermore, the physical-perceptual correspondence was not consistent between warm and cool stimuli, suggesting that warm and cool stimuli have different temporal filtering properties and that cool stimuli are more transient than warm stimuli [12].

Ahmad et al. also found that when multiple temperature stimuli are used to rapidly cool some of the actuators and slowly heat the rest, the slow-heating actuators are not perceived, suggesting that fast temperature changes are perceived at a much lower threshold than slow temperature changes, and that temperature perception is a non-linear phenomenon [13]. Arai et al. found the inverse thermal sensation caused by the presence of hot and cold stimuli, which they called hot-cold confusion [14]. The opposite thermal stimuli applied at multiple locations affect each other, and participants sometimes perceive the hot stimulus at the outer location as cold even when the two of the three stimuli are hot, and vice versa. Several researchers modeled the TGI phenomenon [8,15,16,17,18], though the whole underlying mechanism of the ’grill illusion’ is still ambiguous.

In summary, TGI, thermal referral, and dominance have similar stimulation conditions, though the felt sensations vary. In several thermal referrals and dominance, changes in outer temperature are limited to +10 °C or −10 °C, and the center of the stimuli is set at room temperature. These stimuli induce a similar feeling to that of other fingers, even on room temperature stimulation. On the other hand, under the TGI condition, the adjacent temperature changes to +10 °C and −10 °C simultaneously. The stimuli induce a sensation of pain.

Sato et al. [19] evaluated that spatiotemporal control induces a faster thermal change illusion. In previous studies, thermal stimulation was arrayed in a row (Figure 1a). However, they arrayed the stimulation with the 2 × 2 grid to generate spatially separated stimuli (Figure 1b). They revealed that spatially separated thermal stimuli are perceived as one thermal stimulus and that spatial stimulus change induces faster thermal change perception than a single thermal stimulus change (Figure 2a). On the basis of the result, we proposed employing the spatiotemporal display method to control the thermal illusion. In this research, we treated TGI with our spatiotemporal display method (Figure 1c) and evaluated the effect of the proposed method. With this method, we enhance the effect of the illusion.

Here, we proposed a spatiotemporal control method to realize a variable TGI and evaluated the effect of the method. First, we examined whether there was a change in the period until pain occurred due to the spatial temperature distribution of pre-warming and pre-cooling and verified whether the period until pain occurred became shorter as the temperature difference between pre-warming and pre-cooling increased. Next, we examined the effect of the number of grids and their size on the illusion and verified that increasing the number of grids induces a larger pain sensation and its area simultaneously.

## 2. Spatiotemporal Control on TGI

As mentioned above, spatial distribution using 2 × 2 grids of multiple Peltier elements and its temporal control induced high-speed temperature change sensation (Figure 3a). On the basis of the result, we augmented the control to TGI and evaluated the effect.

### 2.1. Pre-Cooling/Pre-Warming Effect

First, we expanded the spatial control method and controlled the grid-tiled Peltier devices to be pre-cooled and pre-warmed in a checkered pattern simultaneously. We assumed that a rapid temperature change in both the heat sources from the pre-cooling and pre-warming states would cause the perception of pain due to the TGI at a higher speed (Figure 2b,c). The period until pain sensation in our proposed method, $\Delta t^{\prime }$, could be shorter than the period of normal TGI, Δt. Thus, our hypothesis is that a rapid temperature change from pre-cooling and pre-warming states would cause the perception of TGI-based faster pain sensation. In this paper, we evaluated the hypothesis and survey the characteristics of the stimuli.

We used four tiled Peltier devices (Adaptive ET-071-08-15-RS, European Thermodynamics Ltd. (Kibworth Harcourt, UK) 18 mm^2^), combined them with thermistors, and controlled them with a microcomputer. As shown in Figure 3a, we aligned the four Peltier devices and named the upper left and lower right part A and the rest part B. The total size of Peltier devices was 36 mm^2^ We applied warm stimuli in part A and cool stimuli in part B in a checkered pattern. A PC was connected to the microcomputer, and with the PC, we controlled the Peltier devices on the basis of the thermistor input.

The user then placed their finger on the crossing point of the devices (Figure 3b). As an adaptation temperature, we set 33 °C. To adapt to the temperature, the participants kept their hands on the device for three minutes. In the control pattern, as a pre-cooling/pre-warming temperature, we set ±0 °C against the adaptation temperature. In each experimental pattern, as a pre-cooling/pre-warming temperature, we set ±1, 2, and 3 °C against the adaptation temperature. As a cooling/warming temperature target, we set 20 and 40 °C for each. Examples of the thermal histories of parts A and B are shown in Figure 3c.

### 2.2. Spatial Distribution Effect

Second, by designing and controlling the Peltier devices, we examined the spatial distribution effect of the TGI. To evaluate the area effect, we tiled Peltier devices (Adaptive ET-031-10-13-H1, European Thermodynamics Ltd. 15 mm^2^), which was smaller than the previous ones, in 3 × 3 size. The total size of Peltier devices was 45 mm^2^ For 2 × 2 stimuli, the total size is 30 mm^2^ The fingertips were too small to evaluate the spatial effect. Thus, we presented the stimuli to the participants’ forearms. We employed nine Peltier devices, combined them with thermistors, and controlled them with two microcomputers.

As shown in Figure 4a, we aligned the nine Peltier devices and divided the area into A and B. The user then placed their forearm on the devices (Figure 4b). In the control pattern, as a pre-cooling/pre-warming temperature, we set ±0 °C. We prepared 3 × 3 and 2 × 2 stimuli distributions for comparison. For 2 × 2 stimuli, we used four Peltier devices in the lower right, shown in a square dotted in Figure 4c. This time, we prepared four different warm and cool distributions. As an adaptation temperature, we set 33 °C. To adapt to the temperature, the participants kept their hands on the device for three minutes. In the control pattern, as a pre-cooling/pre-warming temperature, we set ±0 °C against the adaptation temperature. In the experimental pattern, as a pre-cooling/pre-warming temperature, we set ±1 and 3 °C against the adaptation temperature. As the target cooling temperature, we set 20 °C. As a target warming temperature, we set 40 °C. With the constructed device, we performed an experiment that evaluated the spatial distribution effect of the TGI.

For each experiment result, to verify the statistical significance between conditions, pairwise comparisons between each condition after one-way repeated measures ANOVA were made using Scheffe’s F-test.

## 3. Experiment

To evaluate the pre-cooling/pre-warming effect and the spatial distribution effect of the TGI, we conducted the following experiments.

### 3.1. Experiment with the Pre-Cooling/Pre-Warming Effect

In this section, to evaluate the pre-cooling/pre-warming effect, we carried out the following psychophysical experiment. In this experiment, we evaluate the periods (Δt in Figure 2b,c) until the pain feeling among the difference between the pre-cooling/pre-warming conditions. Figure 3b shows an overview of the experiment system.

The participants were seven healthy men aged 22 to 24 years. We recruited the participants from our university students. They were paid $10 upon completion of the experiment. During this experiment, they used a noise canceling headphone to block external sounds and listened to pink noise. They were all right-handed and used their right index fingers to feel the stimuli. To eliminate the influence of the order effect, every pattern was presented in random order for each subject.

We compared seven rendering methods of cooling/warming patterns, with ±1, ±2, and ±3/without pre-cooling/pre-warming. They felt the stimuli ten times for each pattern in random order. In ‘without pre-cooling/pre-warming patterns,’ the temperature of part A changes from 33 to 40 °C, and that of part B changes from 33 to 20 °C. In ‘with pre-cooling/pre-warming patterns 1, 2, and 3,’ the temperature of part A had been set at 33 + 1, +2, and +3, and was changed to 40 °C each. According to part A, part B had been set to 33 − 1, −2, and −3 and changed to 20 °C. The temporal patterns are as follows;

±0 for pre-cooling/pre-warming temperature (control)±1 for pre-cooling/pre-warming temperature±2 for pre-cooling/pre-warming temperature±3 for pre-cooling/pre-warming temperature

In all conditions, the participants placed their fingers at the crossing point of the Peltier devices and the cutaneous temperature of their fingertips was controlled to 33 °C by another Peltier device. After that, they pressed the start button of the experiment. When they felt a pain sensation, they were asked to press the button. After each experiment, they answered the 5 stage Likert scale questionnaires about the amount of pain felt (1: No pain, 2: Mild pain, 3: Moderate pain, 4: Severe pain, 5: Intense pain). After the 3 minute intervals, they repeated the experiment 10 times for each condition. In the experiment, the type of stimuli is not announced. To eliminate the sound effect, they listened to pink noise during the experiment.

### 3.2. Questionnaires with Spatial Distribution Effect

In this section, to evaluate the effect of spatial distribution, we conducted the following psychophysical experiment. In this experiment, we compared the spatial distribution effect of the TGI. To compare the effect, we measured the periods (Δt), the area of pain sensation, and the estimate of the magnitude of subjective pain sensation under each condition.

The participants were five healthy men aged 22 to 24 years. We recruited the participants from our university students. They were paid $10 upon completion of the experiment. Participants were partially different from the previous research. During this experiment, they used a noise canceling headphone to block external sounds and listened to pink noise. They were all right-handed and used their right forearms to feel the stimuli. To eliminate the influence of the order effect, every pattern was presented in random order for each subject.

We compared four spatial and three temporal combinations of cooling/warming patterns. The spatial patterns were as follows;

2 × 2 display with warm stimuli in part A, and cool stimuli in part B2 × 2 display with warm stimuli in part B, and cool stimuli in part A3 × 3 display with warm stimuli in part A, and cool stimuli in part B3 × 3 display with warm stimuli in part B, and cool stimuli in part A

The temporal patterns were as follows;

±0 for pre-cooling/pre-warming temperature±1 for pre-cooling/pre-warming temperature±3 for pre-cooling/pre-warming temperature

In all conditions, the participants placed their forearms on the tiled Peltier devices, and the cutaneous temperature of their forearms was controlled to 33 °C by another Peltier device. After that, they pressed the start button of the experiment. When they felt a pain sensation, they were asked to press the button. After each experiment, they asked about the amount of pain felt using the magnitude estimation method, and these data were normalized to 0.0–1.0. In the previous experiment, we used a Likert-scaled questionnaire. This time, to equalize the maximum and minimum responses between participants and to evaluate the pain sensation more precisely, we held a magnitude-estimation-based questionnaire. Using the estimation, we normalized the answers between participants. In addition, they sketched the area of pain in their forearms. The area information was scanned, evaluated, and normalized by the squared thermal area (Figure 4c). After the 3 min intervals, they repeated the experiment 3 times for each condition. In the experiment, the type of stimuli is not announced. To eliminate the sound effect, they listened to pink noise during the experiment.

## 4. Results

Here, we report the results of spatiotemporal control experiments to realize a variable TGI and evaluate the effect of the method. First, we examined whether there was a change in the period until pain occurred due to the spatial temperature distribution and verified whether the period until pain occurred became shorter as the temperature difference between pre-warming and pre-cooling increased. Next, we examined the effect of the number of grids and their size on the illusion and verified that increasing the number of grids induces a larger pain sensation and its area simultaneously.

### 4.1. The Result on the Pre-Cooling/Pre-Warming Effect

Our assumption is that a rapid change in temperature in both the heat sources from the pre-cooling and pre-warming states would cause the perception of pain due to TGI at a higher speed.

Figure 5a shows each period until the participants feel the pain sensation. The x-axis shows each pre-cooling/pre-warming condition, and the y-axis shows periods until the pain sensation. The pastel-colored dashed lines show each participant’s result, and the darkblue line shows the average of all participants’ results. As you can see, there were two outliers in the result. In an interview after the experiment, we found that the two participants were sensitive to cold.

To clarify the pre-cooling/pre-warming effect without the outliers, we excepted the two outliers and plotted the average period again. Figure 5b shows the result. The * and ** indicated a significant difference of p<0.05 and p<0.01 for each. Therefore, there existed a significant difference between the conditions of 33 ± 0 and 33 ± 2, and 33 ± 0 and 33 ± 3.

Furthermore, Figure 5c shows the result of a 5 stage Likert scale questionnaire on the amount of pain felt (1: No pain, 2: Mild pain, 3: Moderate pain, 4: Severe pain, 5: Intense pain). Scheffe’s F-test did not show any significant differences between each pair.

### 4.2. The Result of the Spatial Distribution Effect

To compare the spatial distribution effect of the TGI, we measured the periods (Δt), the area of the pain sensation, and the magnitude estimation of subjective pain sensation under each condition.

#### 4.2.1. Effect on the Periods (Δt) until the Pain Feeling

Figure 6 shows the periods of all participants/conditions until the participants feel the pain sensation. The x-axis shows each combination of spatial distribution and pre-cooling/pre-warming conditions, and the y-axis shows periods until pain sensation. The pastel-colored dashed lines show each participant’s result, and the darkblue line shows the average of all participants’ results. As you can see, there were outliers in the result. In an interview after the experiment, the participant was sensitive to cold.

To clarify the spatiotemporal distribution effect, we excepted for the outlier and plotted the average period again. Figure 7a shows the result of each spatial condition classified by pre-cooling/pre-warming conditiontemporal condition. To examine the temporal effect, we recapped and plotted the data from the temporal point of view (Figure 7b). Therefore, there existed a significant difference between the condition of 33 ± 0 and 33 ± 3. Furthermore, to examine the spatial effect, we recapped and plotted the data from the spatial point of view (Figure 7c). Therefore, there was a significant difference between the conditions of 2 × 2 AwBc—3 × 3 AwBc, 2 × 2 AwBc − 3 × 3 AcBw, 2 × 2 AcBw − 3 × 3 AwBc, and 2 × 2 AcBw − 3 × 3 AcBw. The ‘2 × 2’ and ‘3 × 3’ mean the layout of Peltier device, and the ‘AwBc,’ etc. means the cooling/warming condition of part A and part B. For example, ‘AwBc’ meant that part A was the warming part, and part B was the cooling part.

#### 4.2.2. Effect on the Pain Sensation Area

Figure 8a shows the area result of each spatial condition categorized by pre-cooling/pre-warming conditions. To examine the temporal effect, we recapped and plotted the data from the temporal point of view (Figure 8b). There exists no significant difference between each condition. Furthermore, to examine the spatial effect, we recapped and plotted the data from a spatial point of view (Figure 8c). Thus, there existed a significant difference between the conditions of 2 × 2 AwBc − 3 × 3 AwBc, 2 × 2 AwBc − 3 × 3 AcBw, and 2 × 2 AcBw − 3 × 3 AcBw conditions.

#### 4.2.3. Effect on the Magnitude of the Pain Sensation

Figure 9a shows the magnitude estimation result of each spatial condition categorized by pre-cooling/pre-warming conditiontemporal condition. To examine the temporal effect, we recapped and plotted the data from the temporal point of view (Figure 9b). Thus, there existed a significant difference between the condition of 33 ± 0 and 33 ± 3. Furthermore, to examine the spatial effect, we recapped and plotted the data from a spatial point of view (Figure 9c). Thus, there existed a significant difference between the conditions of 2 × 2 AwBc − 3 × 3 AwBc, 2 × 2 AwBc − 3 × 3 AcBw, 2 × 2 AcBw − 3 × 3 AwBc, and 2 × 2 AcBw − 3 × 3 AcBw.

## 5. Discussion

Here we discuss the results and clarify what is revealed from them.

### 5.1. Discussion on the Pre-Cooling/Pre-Warming Effect

From Figure 5b, we found a significant difference between the conditions of 33 ± 0 and 33 ± 2, and 33 ± 0 and 33 ± 3. The result indicates that pre-cooling/pre-warming induces a shortening of the period, Δt. Furthermore, the larger the pre-cooling/pre-warming temperature, the shorter the period, Δt. From Figure 5c, we could not find significant differences in the amount of pain sensation between each pre-cooling/pre-warming condition.

### 5.2. Discussion on Spatial Distribution Effect

From Figure 7b, we found significant differences between the conditions of 33 ± 0 and 33 ± 3. The result indicates that pre-cooling/pre-warming induced a shortening of the period Δt. Furthermore, the larger the pre-cooling/pre-warming temperature, the shorter the period, Δt. This result is the same as the previous experiment. From Figure 7c, we found significant differences between the conditions of 2 × 2 AwBc − 3 × 3 AwBc, 2 × 2 AwBc − 3 × 3 AcBw, 2 × 2 AcBw − 3 × 3 AwBc, and 2 × 2 AcBw − 3 × 3 AcBw. The result indicates that the larger thermal area induces a shortening of the period, Δt. On the other hand, we found no significant differences between the conditions of 2 × 2 AwBc − 2 × 2 AcBw, and 3 × 3 AwBc − 3 × 3 AcBw. This indicated that the layout of the cooling/warming position does not affect the period.

From Figure 8b, we found no significant differences between each condition. The result indicates that pre-cooling/pre-warming does not affect the pain area. From Figure 8c, we found significant differences between the conditions of 2 × 2 AwBc − 3 × 3 AwBc, 2 × 2 AwBc − 3 × 3 AcBw, 2 × 2 AcBw − 3 × 3 AwBc, and 2 × 2 AcBw − 3 × 3 AcBw. The result indicated that the larger thermal area increased the pain area. On the other hand, we found no significant differences between the conditions of 2 × 2 AwBc − 2 × 2 AcBw, 3 × 3 AwBc − 3 × 3 AcBw. This indicated that the layout of the cooling/warming position does not affect the pain area.

From Figure 9b, we found no significant differences between each condition. The result indicates that pre-cooling/pre-warming did not affect the magnitude of the pain sensation. From Figure 9c, we found significant differences between the conditions of 2 × 2 AwBc − 3 × 3 AwBc, 2 × 2 AwBc − 3 × 3 AcBw, 2 × 2 AcBw − 3 × 3 AwBc, and 2 × 2 AcBw − 3 × 3 AcBw. The result indicated that the larger thermal area induces a magnifying of the magnitude of the pain sensation. On the other hand, we found no significant differences between the conditions of 2 × 2 AwBc − 2 × 2 AcBw, 3 × 3 AwBc − 3 × 3 AcBw. This indicated that the layout of the cooling/warming position did not affect the magnitude of the pain sensation.

From Figure 8c and Figure 9c, we found that the pain areas by the stimulus of 3 × 3 were twice larger than 2 × 2. The amount of pain sensation by the stimulus of 3 × 3 was also twice larger than 2 × 2. Here, the device’s area of 3 × 3 is 2.25 times larger than 2 × 2. Thus, the reason could be the difference of area. However, Figure 8c also shows that everyone felt a pain sensation in a part of the thermal area. The drawings of the pain area by the participants also show the same result. That is, it is possible that pain sensation is not induced by stimulation of the entire thermal area, but rather its borders. In the near future, we want to consider the causes in a more profound way. Besides, the epidermis of the male is thicker than the female. In our experiment, the participants were all male. When conducting the same experiment with the female, the results might be different. In the near future, we want to hold the experiment with female participants.

## 6. Conclusions

In this paper, we proposed a spatiotemporal control method to realize the variable TGI and evaluated the effect of the method. First, we examined whether there was a change in the period until pain occurred due to the spatial temperature distribution of pre-warming/pre-cooling and verified whether the period until pain occurred became shorter as the temperature difference between pre-warming/pre-cooling increased. Then, we examined the effect of the number of grids and their size on the illusion and verified that increasing the number of grids induces a greater pain sensation and its area simultaneously.

To evaluate the spatiotemporal control on TGI, we performed two experiments on the pre-cooling/pre-warming effect and the spatial distribution effect. Through the experiments, we found that our hypothesis was correct. Thus, in terms of the period, Δt, pre-cooling/pre-warming induced a shortening of the period. Furthermore, the larger the pre-cooling/pre-warming temperature, the shorter the period. Furthermore, the larger thermal area induced a shortening of the period.

In terms of the pain area, the larger thermal area increased the pain area. Pre-cooling/pre-warming did not affect the pain area. In terms of the pain magnitude, the larger thermal area magnified the magnitude of the pain sensation. Pre-cooling/pre-warming did not affect the pain magnitude. The amount of pain caused by the stimulus of 3 × 3 was twice greater than that of 2 × 2. The pain area of 3 × 3 was also twice larger than 2 × 2. Here, the device’s area of 3 × 3 is 2.25 times larger than 2 × 2. Thus, the reason could be the difference of the area. However, everyone felt a pain sensation in a part of the cooling/warming area. That is, it was possible that the sensation of pain was not induced by stimulation of the whole thermal area, but rather its borders. In the near future, we want to consider the causes more deeply. Furthermore, the layout of the cooling/warming position did not affect the period, the area of pain, and the sensation of pain.

From the findings of our research, we have got a basic design method for the pain display. By controlling the thermal device spatially and temporally, we can control the amount of pain sensation and pain area without any physical damage. Pre-warming and pre-cooling temperature control induces temporal changes until the sensation of pain. To cause the wider pain sensation area, the stimulation area control is effective. To induce the pain sensation more strongly, the stimulation area or pre-warming and pre-cooling temperature control is effective. In the future, to reveal its mechanism, we will carry out deeper spatial/temporal research.

## Figures and Tables

**Figure 1 sensors-23-00414-f001:**
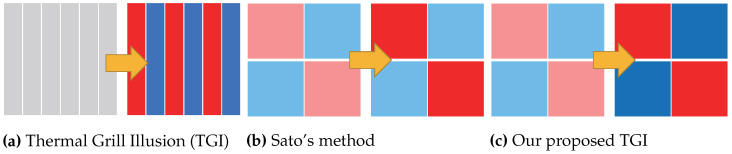
Thermal change patterns of each method. (**a**) Thermal stimulation on TGI. The stimulation was arrayed in a row. (**b**) Thermal stimulation on Sato’s method. Peltier elements are grid-arrayed, and a part of the thermal pattern is changed. (**c**) Thermal stimulation on the proposed method. Peltier elements are grid-arrayed, and both parts of the thermal pattern are changed.

**Figure 2 sensors-23-00414-f002:**
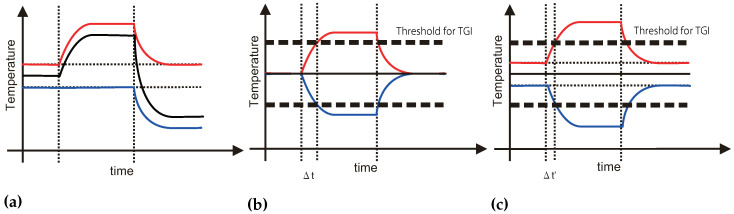
Modelized thermal histories of the conventional and proposed method. (**a**) Thermal history of Sato’s method. The red and blue line shows stimulated temperature on each grid-arrayed Peltier element. The solid black line shows the perceived temperature. The spatially separated pre-warmed or pre-cooled thermal stimulation induces faster recognition of thermal change. (**b**) Thermal history of normal TGI stimulation. The red and blue line shows stimulated temperature on arrayed Peltier elements in a row. The solid black line shows the perceived temperature. The thick dotted line shows the threshold of the pain sensation. Δt shows the period until pain sensation. (**c**) Thermal history of the proposed method. The spatially separated pre-warmed and pre-cooled thermal stimulation could induce faster recognition of the pain sensation. $\Delta t^{\prime }$ shows the period until pain sensation.

**Figure 3 sensors-23-00414-f003:**
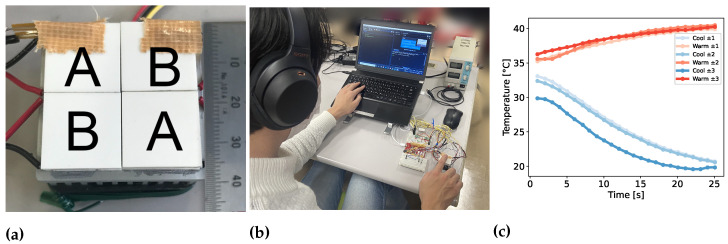
Experiment setup of pre-cooling/pre-warming effect. (**a**) 2 × 2 tiled Peltier devices. The total size of Peltier devices was 36 mm${}^{2}$ (**b**) System overview of the experiment about pre-cooling/pre-warming effect (**c**) Examples of both A and B parts’ thermal histories. As an adaptation temperature, we set 33 °C. To adapt to the temperature, the participants kept their hands on the device for three minutes. As a cooling/warming temperature target, we set 20 and 40 °C for each.

**Figure 4 sensors-23-00414-f004:**
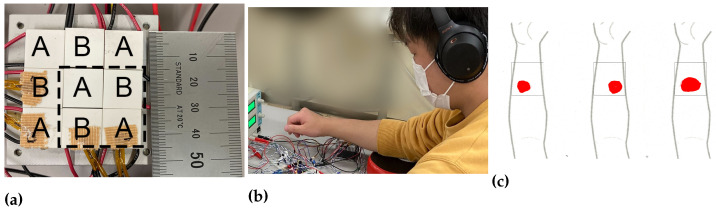
Experiment setup of spatial distribution effect. (**a**) 3 × 3 tiled Peltier devices. The total size of Peltier devices was 45 mm${}^{2}$ For 2 × 2 stimuli, the total size was 30 mm${}^{2}$ (**b**) System overview of the experiment about spatial distribution effect. As an adaptation temperature, we set 33 °C. To adapt to the temperature, the participants kept their hands on the device for three minutes. As a cooling/warming temperature target, we set 20 and 40 °C for each. (**c**) Participant’s drawing examples of the pain area.

**Figure 5 sensors-23-00414-f005:**
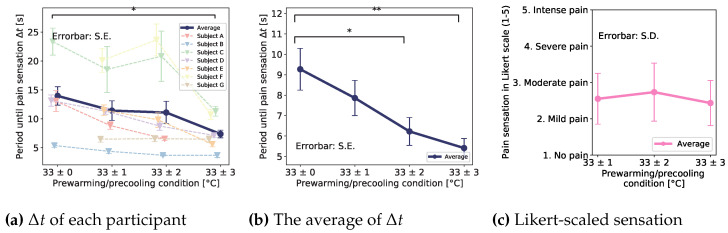
Results of pre-cooling/pre-warming effect. (**a**) Each participant’s period until pain sensation and their average (**b**) The period of average and significant difference (except the outlier) (**c**) An average of 5 stages Likert-scaled questionnaires about the sensation. The * and ** indicated a significant difference of p<0.05 and p<0.01 for each.

**Figure 6 sensors-23-00414-f006:**
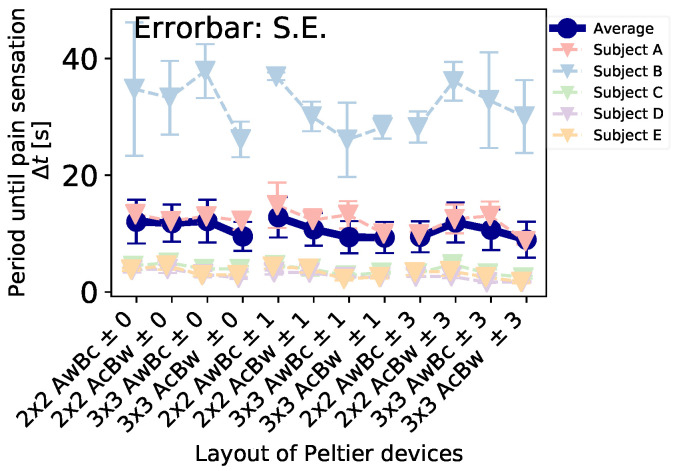
Δt’s result of spatial distribution effect.

**Figure 7 sensors-23-00414-f007:**
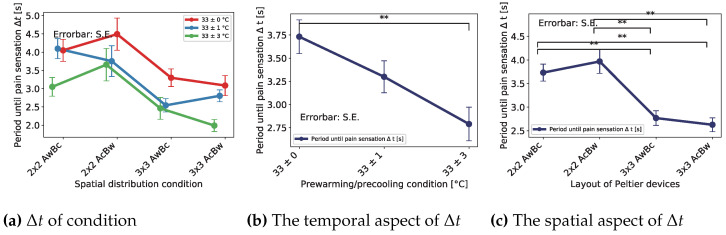
Δt’s results of spatial distribution effect (except the outlier). The ** indicates a significant difference of p<0.01. (**a**) Effect of layout of Peltier device and pre-cooling/pre-warming condition on time until pain. (**b**) Averaged effect of pre-cooling/pre-warming condition on time until pain. (**c**) Averaged effect of layout of Peltier device on time until pain.

**Figure 8 sensors-23-00414-f008:**
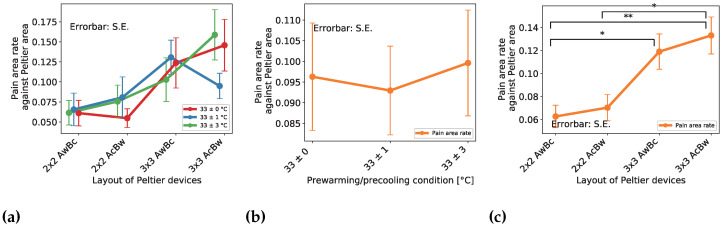
Area of the pain sensation result of spatial distribution effect. The * and ** indicates a significant difference of p<0.05 and p<0.01 for each. (**a**) Effect of layout of Peltier device and pre-cooling/pre-warming condition on pain area rate (**b**) Averaged effect of pre-cooling/pre-warming condition on pain area rate (**c**) Averaged effect of layout of Peltier device on pain area rate.

**Figure 9 sensors-23-00414-f009:**
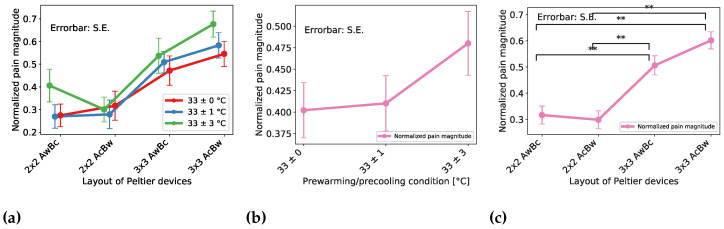
Magnitude estimation of the pain sensation result of spatial distribution effect. The ** indicates a significant difference of p<0.01. (**a**) Effect of layout of Peltier device and pre-cooling/pre-warming condition on magnitude estimation rate (**b**) Averaged effect of pre-cooling/pre-warming condition on magnitude estimation rate (**c**) Averaged effect of layout of Peltier device on magnitude estimation rate.

## Data Availability

Data sharing not applicable because of the ethical issues.

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
