# Peer review of "Spatiotemporal Thermal Control Effects on Thermal Grill Illusion"

_sensors, 2022, doi:10.3390/s23010414_

Round 1

Reviewer 1 Report

 I am not an expert in this field, but found this work is pretty interest that spatiotemporal thermal control effects on thermal grill illusion. The conclusion is too long, need to be improved.  And the contents of figure shoule be more clear, some information overlaps in Figure 5.

  •  

  •  

Author Response

Response to reviewer #1

We would first like to thank all the reviewers and editors for their time and constructive comments! Please find our answer to the reviewer's questions below.

Comments and Suggestions of Reviewer #1

 I am not an expert in this field, but found this work is pretty interest that spatiotemporal thermal control effects on thermal grill illusion. The conclusion is too long, need to be improved.  And the contents of figure shoule be more clear, some information overlaps in Figure 5.

Thank you for your suggestion. We reconsider the construction of the paper and append

several detailed descriptions to each part. Please check them.

  •  I am not an expert in this field, but found this work is pretty interest that spatiotemporal thermal control effects on thermal grill illusion. The conclusion is too long, need to be improved.  And the contents of figure shoule be more clear, some information overlaps in Figure 5.
      • Thank you for your comment. I rewrite Fig. 5.
      • I simplified the conclusion in terms of the following 3 points. 
  • To evaluate the spatiotemporal control on TGI, we performed the following two experiments about the pre-cooling / pre-warming effect and the spatial distribution effect. Through the experiments, we found that our hypothesis is correct. Thus, in terms of the period, $\Delta t$, pre-cooling / pre-warming induced a shortening of the period. Furthermore, the larger the pre-cooling / pre-warming temperature, the shorter the period. Furthermore, the larger thermal area induced a shortening of the period. 
  • In terms of the pain area, the larger thermal area increased the pain area. Pre-cooling / pre-warming did not affect the pain area.  In terms of the pain magnitude, the larger thermal area magnified the magnitude of pain sensation. Pre-cooling / pre-warming did not affect the pain magnitude. The amount of pain caused by the stimulus of 3x3 was twice greater than that of 2x2. The pain area of 3x3 was also twice larger than 2x2. Here, the device's area of 3x3 is 2.25 times larger than 2x2. Thus, the reason could be the difference of area. However, everyone felt a pain sensation in a part of the cooling / warming area. That is, it was possible that the sensation of pain was not induced by stimulation of the whole thermal area, but rather its borders. In the near future, we want to consider the causes more deeply.  In addition, the layout of the cooling/warming position did not affect the period, the area of pain, and the sensation of pain.

Reviewer 2 Report

1. Please clarify the reason of participants selection: a) why uses male only and why chooses the age range of 22-24? b) are participants of different sections the same people or different or partial different? 

2. The result is based on the feelings of participants, however, the pain feelings vary. Please discuss whether or how the different participants affect your result/accuracy.

3. The abstract part needs more information

Author Response

Response to reviewer #2

We would first like to thank all the reviewers and editors for their time and constructive comments! Please find our answer to the reviewer's questions below.

Comments and Suggestions of Reviewer #2

  1. Please clarify the reason of participants selection: a) why uses male only and why chooses the age range of 22-24? b) are participants of different sections the same people or different or partial different? 
  2. The result is based on the feelings of participants, however, the pain feelings vary. Please discuss whether or how the different participants affect your result/accuracy.
  3. The abstract part needs more information

Thank you for your suggestion. We reconsider the construction of the paper and append

several detailed descriptions to each part. Please check them.

  • Please clarify the reason of participants selection: a) why uses male only and why chooses the age range of 22-24? b) are participants of different sections the same people or different or partial different? 
      • Thank you for your comment. I appended the following explanation of participants in the experiment section, and the discussion section.
  • Section 3.1
  • We recruited the participants from our university student. They were paid \$10 upon completion of the experiment.
  • Section 3.2
  • We recruited the participants from our university student. They were paid \$10 upon completion of the experiment. The participants were partially different from the previous research.
  • Section 5. Discussion
  • Besides, the epidermis of male is thicker than female. In our experiment, the participants were all male. By holding the same experiment with female, the results might be different. In the near future, we want to hold the experiment female participants. 
  • The result is based on the feelings of participants, however, the pain feelings vary. Please discuss whether or how the different participants affect your result/accuracy.
      • As the reviewer says, the pain feeling vary between participants. Though, in the pre-cooling/pre-warming experiment, we held Likert-scale-based questionnaire. From the result, we got relatively small standard deviation (1.0 or little over). This indicated that we got stable result from the questionnaire. Furthermore, to equalize the maximum and minimum answers between participants and to evaluate the pain sensation more precisely, in the following experiment we held magnitude estimation based questionnaire. By employing the estimation, we normalized the answers between participants. To clarify the purpose, we appended the following description. 
  • In section 3.2
  • In the previous experiment, we used Likert-scaled questionnaire, though, to equalize the maximum and minimum answers between participants and to evaluate the pain sensation more precisely, in the following experiment we held magnitude estimation based questionnaire. By employing the estimation, we normalized the answers between participants.
  • The abstract part needs more information
      • Thank you for your comment. To clarify what is TGI and what we contributed, we rewrote the abstract as follows;
  • The thermal grill illusion induces the pain sensation induces a burning sensation under a spatial display of warmth and coolness of approximately 40 °C and 20 °C. To realize virtual pain display more universally during the virtual reality experience, we proposed a spatiotemporal control method to realize a variable thermal grill illusion and evaluated the effect of the method. First, we examined whether there was a change in the period until pain occurred due to the spatial temperature distribution of pre-warming and pre-cooling and verified whether the period until pain occurred became shorter as the temperature difference between pre-warming and pre-cooling increased. Next, we examined the effect of the number of grids on the illusion, and verified the following facts. In terms of the pain area, the larger thermal area increased the pain area. In terms of the pain magnitude, the larger thermal area magnified the magnitude of pain sensation.
